# Homeostatic Regulation of the Duox-ROS Defense System: Revelations Based on the Diversity of Gut Bacteria in Silkworms (*Bombyx mori*)

**DOI:** 10.3390/ijms241612731

**Published:** 2023-08-12

**Authors:** Qilong Shu, Xiqian Guo, Chao Tian, Yuanfei Wang, Xiaoxia Zhang, Jialu Cheng, Fanchi Li, Bing Li

**Affiliations:** 1School of Basic Medicine and Biological Sciences, Soochow University, Suzhou 215123, China; 18985575232@163.com (Q.S.); m13345325483@163.com (X.G.); tc17683956092@163.com (C.T.); wangyuanfeime@163.com (Y.W.); 13541770069@163.com (X.Z.); 18052861838@163.com (J.C.); fcli@suda.edu.cn (F.L.); 2Sericulture Institute, Soochow University, Suzhou 215123, China

**Keywords:** Duox-ROS, silkworm, 16S rRNA, intestinal microbiota

## Abstract

The Duox-ROS defense system plays an important role in insect intestinal immunity. To investigate the role of intestinal microbiota in Duox-ROS regulation herein, 16S rRNA sequencing technology was utilized to compare the characteristics of bacterial populations in the midgut of silkworm after different time-periods of treatment with three feeding methods: 1–4 instars artificial diet (AD), 1–4 instars mulberry leaf (ML) and 1–3 instars artificial diet + 4 instar mulberry leaf (TM). The results revealed simple intestinal microbiota in the AD group whilst microbiota were abundant and variable in the ML and TM silkworms. By analyzing the relationship among intestinal pH, reactive oxygen species (ROS) content and microorganism composition, it was identified that an acidic intestinal environment inhibited the growth of intestinal microbiota of silkworms, observed concurrently with low ROS content and a high activity of antioxidant enzymes (SOD, TPX, CAT). Gene expression associated with the Duox-ROS defense system was detected using RT-qPCR and identified to be low in the AD group and significantly higher in the TM group of silkworms. This study provides a new reference for the future improvement of the artificial diet feeding of silkworm and a systematic indicator for the further study of the relationship between changes in the intestinal environment and intestinal microbiota balance caused by dietary alterations.

## 1. Introduction

The insect gut possesses a complex microbial community and these microbial symbionts have been revealed to be involved in a variety of physiological activities, including nutrition and digestion [1,2], reproductive development [3,4], drug resistance [5], immune defense [6] and the degradation of harmful substances [7]. Under normal physiological conditions, the intestinal microbial population exhibits a stable fluctuation range. Drastic changes result in an imbalance in the microbial homeostasis of the microbiota, directly affecting the growth, development, reproduction and other biological activities of insects [8,9]. Studies have shown a decrease in the intestinal microbiota diversity and richness of silkworms fed with lettuce, disrupting the healthy intestinal microflora’s homeostasis, leading to a reduced digestive enzyme activity and hindering the growth of silkworms [10]. By adding low doses of TiO_2_ NPs, the abundance of individual species of intestinal microorganisms in silkworms is altered, the abundance of *Lachnospiraceae_NK4A136*_group, involved in the metabolism of nutrients, as well as the abundance of *Pseudomonas* and *Sphingomonas*, both involved in detoxification and disease resistance, increased, which promotes the growth and development of silkworms and improves their immune system [11].

The regulation of intestinal microbiota is regulated by several factors, such as intestinal pH [12], digestive enzymes [13], redox potential [14] and the immune system [15]. It has previously been demonstrated that the dual oxidase Duox-mediated production of reactive oxygen species (ROS) is the primary immune mechanism regulating gut–microbial homeostasis in insects [16,17], and that pathogenic microorganisms activate the intestinal Duox immune defense system to produce ROS that directly destroy pathogenic bacteria, fungi and plasmodium [18,19]. The silencing of the *Duox* gene in the intestine of *Bactrocera dorsalis* led to a significant increase in the abundance of intestinal bacteria and a significant change in the structure of the bacterial community, leading to disorders of the intestinal microbiota [20]. *Drosophila melanogaster* and *Aedes aegypti* regulate *Duox* gene expression, further leading to changes in intestinal ROS levels, ultimately regulating the proliferation of intestinal commensal bacteria [21]. In addition to participating in host immune defense, ROS were revealed to alter the permeability of the intestinal mucosa through the action of immunostimulants and block the intestinal overactive immune response. In addition, Duox-generated ROS act as signaling molecules to induce repair responses or the initiation of other homeostatic signaling pathways [22,23]. Therefore, the basal expression and activation of the Duox-ROS defense system is essential for symbiotic microbiota in insect gut health.

As a model insect of the order Lepidoptera, the gut of the silkworm (*Bombyx mori*) is enriched with microflora and is often employed in the study of the relationship between microorganisms and hosts [24]. As an economically important insect, the silkworm can be fed an artificial diet at all instars, but there are associated problems such as small body size and a low larval survival rate in the AD-fed silkworm, and the mechanism of these changes not fully understood [25,26]. A recent study reported that at first–third instars-fed artificial feed, intestinal microbiota associated with instar growth exhibited a single trend and the intestinal pH was acidic, whilst the traditional mulberry-leaf-fed silkworm’s intestine is alkaline [27,28]. In this study, the gut microbial composition of the midgut of silkworms was characterized at different timepoints after being fed artificial diets (ADs), mulberry leaf at 1–4 instars (ML) and artificial feeding at 1–3 instars, followed by mulberry leaf feeding during the fourth instar (TM). Analysis was performed using 16S rRNA sequencing technology to investigate the interactions between food–gut microbiota and the Duox defense system to study the linkage between the gut microbiota diversity and the Duox defense system of the silkworm (Figure 1).

## 2. Result

### 2.1. Estimation of Gut Microbial Diversity and Abundance

In order to compare the differences in gut microbial structure among silkworm in the AD, ML and TM groups, 16S rRNA gene sequencing was performed on samples of the midgut taken from the three groups. As shown in Figure 2, α-diversity indices (Shannon, Simpson, Chao1 and Ace) were used to evaluate the species diversity of individual samples (Appendix A). The results revealed that at 24 h and 48 h, the Shannon index of the AD group was lower than that of the ML and TM groups, and the Simpson index was higher in the AD group than in the ML and TM groups (Figure 2A,B); both of these indices indicate that the diversity of the microbiota of the AD group was lower than that of the ML and TM groups. Moreover, at 24 h, the Ace index of the AD and TM groups was significantly different from that of the ML group. However, the Ace index and Chao1 index of the AD and TM groups were not significantly different at 24 h, indicating that the richness of the microbiota was lower for these groups than that of mulberry-leaf-nursery-fed silkworms (Figure 2C,D). Furthermore, at 48 h, the Ace index and Chao1 index of the AD and TM groups were once again significantly different from the ML group, indicating that there were differences in the richness upon comparing the microbiota of the three feeding groups.

### 2.2. Intestinal Microbial Composition

This study evaluated the 16S rRNA sequencing results of microorganisms isolated from silkworm midgut samples. The sequencing depth enabled the comparison of diversity indices between samples, and the sparsity curves were balanced after removing a large number of chloroplast sequences from the raw sequencing data (Appendix A). The OTU Venn diagram analysis revealed that the AD group exhibited the lowest number of OTUs and demonstrated a decreasing trend, and the three treatment groups had a common number of OTUs in 24 h and 48 h of 11 (Figure 3A). By analyzing the phylum levels of microorganisms in the three treatment groups, the AD group samples consisted primarily of the phylum *Firmicutes*, which increased from 85.27% to 97.86% at 48 h. Meanwhile, samples from the ML and TM groups mainly consisted of the phyla *Proteobacteria*, *Actinobacteria* and *Firmicutes*; bacterial composition at the phylum level in the ML group was relatively stable, but the *Firmicutes* in the TM group increased from 15.11% to 38.09% at 48 h (Figure 3B).

The structure of the intestinal microbiota was compared between groups using multivariate statistical analysis. The results of the principal co-ordinates analysis (PCoA) revealed that the intestinal microbiota of the AD group was separated with the ML and TM groups, and there was a significant bacterial structure. There was no significant difference observed when comparing the microbiota in the TM and ML groups at 24 h, while there was a trend of separation identified at 48 h (Figure 3C). On the basis of this finding, non-metric multidimensional scaling (NMDS) analysis was verified using the Bray–Curtis similarity method, and the results were concordant (Figure 3D).

The distribution of intestinal microorganisms, reared using distinct feeding methods, at the genus level was further analyzed using a community heat map (Figure 4A). It was further identified that the AD group had a primarily monocultural microbiota, with *Pediococcus* and *Weissella* as the main genera; these were less present in the ML and TM groups, with *Pediococcus* accounting for more than 60% at 24 h and 48 h in the AD group, and *Weissella* rising from 20.76% at 24 h to 35.34% at 48 h (Appendix A). The ML and TM groups exhibited the similar genera, *Acinetobacter* and *Gordonia*; the ML group had an abundance of *Methylobacterium-Methylorubrum* and *Sphingomonas*, whilst *Bacillus* and *Curtobacterium* were abundant in the TM group. In this study, we compared the differences between TM and ML groups at 24 h and 48 h, respectively, and observed that there were significant differences in the levels of *Brevundimonas* at 24 h (Figure 4B) and of *Methylobacterium-Methylorubrum*, *Sphingomonas*, *Lysobacter* and *Ileibacterium* at 48 h (Figure 4C).

### 2.3. Expression of Genes Related to the Intestinal Duox-ROS Immune System and Determination of Intestinal Antioxidant Enzyme Activity

ROS levels were determined by measuring ROS levels in the silkworm treated in three groups. At 24 h, the highest ROS level was the larvae of the ML group and this was significantly different from the ROS levels observed in AD and TM groups. Moreover, at 48 h, the ROS level in the TM group increased to be similar to that of the ML group, which was 2.62 times higher than at 24 h. The ROS level in the AD group revealed a decreasing trend over time (Figure 5A) and was consistently the lowest of the three treatment groups.

Based on the observations of ROS level alterations, we employed RT-qPCR to detect the expression levels of Duox-ROS system-related genes. The ML group was used as the control group, and at 24 h, the transcription level of the *Mesh* gene was revealed to be lower in the AD group than in the ML group, whilst it was significantly upregulated 2.39-fold in the TM group compared to the ML group; the transcription level of the *Arrestin* gene was lower in the AD group than in the ML and TM groups, and was significantly upregulated in the TM group compared to the ML group. Moreover, the transcription level of the *Duox* gene was highest in the AD group, 4.02-fold higher than in the ML group. The *Duox* gene in the TM group was similar to that in the ML group at the transcriptional level (Figure 5B–D). At 48 h, Mesh transcript level was the lowest in the AD group, but not significantly different from that in the ML group; however, it was significantly upregulated in the TM group compared to the two other groups. The transcript levels of *Arrestin* gene and *Duox* gene transcript levels were lowest in the AD group, and were significantly different from those in the ML and TM groups. Furthermore, *Arrestin* and *Duox* transcript levels were identified to be significantly upregulated in the TM group. 

To investigate the level of antioxidant enzyme activity in the midgut of the silkworm after changing from feed to mulberry leaves, the antioxidant enzyme activities (SOD, TPX, CAT) in the midgut were measured (Figure 5E–G). It was found that the SOD enzyme activity in the AD group was the highest at both 24 h and 48 h. Meanwhile, the TM group exhibited the lowest level of SOD enzyme activity, whilst the SOD enzyme activity of the ML group decreased with time. Furthermore, TPX enzyme activity in the ML group was the highest at the 24 h timepoint, and the TPX activity in the AD group was the lowest, with a significant difference between the two. However, at 48 h, the TPX activity in the ML and TM groups was decreased, and the enzyme activity of the AD group was the highest, which was 4.29-fold and 7.51-fold of the ML and TM groups, respectively. CAT enzyme activity was the highest in the AD group at both 24 h and 48 h, while the CAT activity in the TM group exhibited an increasing trend compared to a decreasing trend observed in the ML group.

### 2.4. Influence of Pathogenic Bacteria on Body Weight and Organismal ROS Levels in the Silkworm

In order to verify the regulatory effect of bacterial diversity in ROS generation and regulation, the silkworms were inoculated with *E. cloacae* on day 2 of the fourth instar in this study, and upon 24 h after inoculation, no significant difference was observed from the measured body weight compared to that of the relevant control group (Figure 6A). Furthermore, the ROS level was significantly increased in the inoculated silkworm as identified by measuring the ROS level in the midgut (Figure 6B). The level of ROS in the AD control group was 168.20, in the AD test group, 388.29 and in the TM test group, it was 2.77-fold higher than that in the TM group, indicating that the pathogenic bacteria resulted in the accumulation of ROS in the gut of silkworms to facilitate protection against the invasion of the foreign pathogenic bacteria.

## 3. Discussion

### 3.1. Changes in Diet Alter the Composition in the Gut Microbiota of the Traditional Silkworm

The intestinal microbiota of silkworms in the AD group were simpler, and after switching from being artificial feed-fed to mulberry leaf-fed, a significant increase in the abundance and diversity of intestinal microorganisms was observed in the larvae of the TM group, albeit this did not quite reach parity with the silkworms fed with mulberry leaf for all instars (Figure 4A). By measuring intestinal pH, it was identified that the pH of the AD group gradually decreased, whilst the intestine of the ML group remained at around pH 9.40 and the intestinal pH of the TM group gradually increased (Appendix A). It was hypothesized that an acidic pH environment would inhibit the growth of dominant beneficial bacteria in the silkworm, leading to a significant decrease in the diversity of intestinal microorganisms and increasing the possibility that the structure of the microbiota would be destroyed and host resistance would become weaker, thus increasing the risk of disease transmission. An analysis of the top six most abundant KEGG pathway level 1 pathways using a heat map (Tax4Fun functional prediction analysis) revealed the lowest metabolic levels and lowest body weight in the AD silkworm group, and the TM group had significant nutritional compensation, with body weight gain similar to the ML group at 48 h (Appendix A). The simple intestinal microbiota in the AD silkworm group may be the primary reason for stunted growth and a reduced resistance.

At 24 h, there was no significant difference identified when comparing the ROS level in the midgut of the TM group to in the AD group, but there was a significant difference in the activity of antioxidant enzymes (SOD, TPX, CAT). These findings suggest that there was an adaptation process in the midgut of silkworms in the TM group, and the antioxidant enzymes of the silkworms were maintained within a certain range that did not inhibit the production of ROS [21,29]. According to the analysis of differences between groups (Figure 4B), it was observed that there were significant differences between the ML and TM groups at 24 h for *Brevundimonas*, and at 48 h for *Methylobacterium-Methylorubrum*, *Sphingomonas*, *Ileibacterium* and *Lysobacter*. There is a direct relationship between the intestinal pH and the microbiota of the silkworm; the pH of the TM group at both 24 h and 48 h was lower than that of the ML group, and there was a significant difference in gut microbiota, while the ML intestinal pH was stable and no significant difference was found upon comparing the intestinal microbiota of the ML group at 24 h and 48 h (Appendix A).

### 3.2. Enriched Intestinal Microbiota Induces Activation of the Duox-ROS Immune Defense Pathway

DUOX-mediated ROS production in the intestine is regulated via two primary signaling pathways: DUOX enzymatic activity is regulated by the Gαq-PLCβ-Ca^2+^ signaling pathway, and *Duox* gene expression is regulated by the MEKK1-MKK3-p38-ATF2 signaling pathway [30]. Bacterial peptidoglycan PGN can activate the *Duox* gene expression signaling pathway, but not DUOX enzymatic activity. Peptidoglycan alone cannot induce DUOX-mediated ROS production, and as such, the non-peptidoglycan ligand uracil secreted by bacteria is further required to activate Duox-mediated ROS production [31]. However, only foreign microbiota can secrete uracil, and long-term coevolution has resulted in the loss of host intestinal commensal bacteria’s ability to secrete uracil [32]. In this study, we identified that the transcript levels of Duox-ROS immune system-related genes and the levels of ROS in the AD group were significantly lower than in the ML and TM groups. Meanwhile, the transcript levels of Duox-ROS immune system-related genes in the TM group were significantly upregulated accordingly and the levels of ROS were elevated to be similar to those in the ML group (Figure 5). These findings may be a result of the autoclaving of the artificial feed, resulting in a simple intestinal microbiota in the AD group. Without the stimulation of foreign bacteria, a small amount of ROS was generated and a lack of basal ROS resulted in the overgrowth of the host intestinal symbiotic bacteria. Symbiotic bacteria cannot induce the activation of the DUOX immune defense system, which leads to a deficiency in ROS available to participate in homeostatic regulation of the structural composition of the intestinal microbiota in the silkworm. This homeostatic imbalance further results in the stunted growth and development of the silkworm. By infecting silkworms with *E. cloacae* after 48 h of fourth instar, it was revealed that at 72 h, ROS levels were significantly increased, and the ROS level in the AD control group also increased by about 4.62-fold at 48 h (Figure 6B), which is in line with the study of Qin et al., who demonstrated that prolonged exposure to artificial feed attracts bacteria in the environment for mass multiplication. Together with the crawling and feeding behavior of silkworms and the common fecal fermentation, this can lead to the colonization of *Lactobacillus* and *Weissella*, further leading to the acidification of the intestinal environment, and eventually producing pathogenic bacteria [33]. At 72 h, ROS in the AD control silkworm was significantly upregulated due to stimulation to combat its own microbiota imbalance and exposure to pathogenic bacteria. Overall, these factors stimulate the accumulation of ROS in the AD silkworm group.

### 3.3. Distribution of Key Dominant Genera in the Intestinal Microbiota of the Silkworm

Distinct from the traditional mulberry leaf, the artificial feed contains no foreign microbiota, resulting in a monoculture of intestinal microbiota in silkworms, with only *Pediococcus* and *Weissella* present. *Pediococcus*, a form of lactic acid-producing bacteria, are an important species in maintaining immune regulation and antioxidant activity in the gut, and some studies have reported that this genus can produce bacteriocins, which have an inhibitory effect on foreign bacteria [34,35]. Furthermore, *Weissella* are capable of producing lactic acid, facilitating the fermentation of plant foods in the animal intestine and promoting intestinal absorption [36,37]. As such, *Pediococcus* and *Weissella* are often used as probiotic additives in animal feed.

The silkworm is in a state of nutritional compensation after the switch to mulberry leaf from an artificial diet, and therefore has a requirement to absorb energy through growth, wherein its gut microbes aid in digestion and utilization [38]. *Bacillus* and *Lysobacter* were present in greater numbers in the TM group at 48 h compared to 24 h (Appendix A). Notably, *Bacillus* possesses the ability to degrade plant cellulose and produce proteases, lipases and cellulases to aid nutritional compensation in the silkworm [39,40]. Moreover, *Lysobacter* is an important biocontrol bacterium that produces antibiotics against plant pathogens and is capable of producing non-volatile antimicrobial metabolites [41]. In the TM group, after exposure to mulberry leaves, the colonies present may cause the organism to undergo a period of adaptation, wherein the gut immune system of the silkworm will screen for beneficial bacteria and stabilize its intestinal microbiota. The combined effect of the organism’s own immunity and the intestinal microbiota inhibits the presence of harmful pathogens, and as such, a difference in microbiota between 24 h and 48 h is observed in the TM group. Conversely, there is no significant difference between 24 h and 48 h in the ML group, indicating a stable intestinal microbiota.

Moreover, *Methylobacterium-Methylorubrum*, *Acinetobacter* and *Sphingomonas* were present in the ML group (Figure 4A). Studies have demonstrated that *Methylobacterium-Methylorubrum* is involved in nitrogen fixation and cellulose degradation in silkworm tissues [42] and *Acinetobacter* exhibits starcholytic, cellulolytic, xylanolytic and esterase activities [43,44]. *Sphingomonas* was identified to be one of the genera of intestinal cellulolytic bacteria in the Chinese white pine beetle, *Dendroctonus Armandi larvae* [45]; it has been shown to efficaciously degrade organic matter [46] and is associated with the detoxification of the female kale beetle *Colaphellus bowringi* [47]. Therefore, the dominant colonies present in the ML group are involved in the digestion and absorption of nutrients in the silkworm.

## 4. Materials and Methods

### 4.1. Silkworm Strains and Rearing

Silkworms of the strain *B. mori* (Youshi-1) were provided by Shandong Guangtong Silkworm Seed Group Co., Ltd. (Weifang, China). The artificial diet was purchased from Chongqing Zhengjia Feedstuff Co., Ltd. (Chongqing, China). The artificial diet consisted of 40% mulberry leaf powder, 30% soybean meal, 25% corn starch, 2.5% multivitamins, 1.5% inorganic salt and 0.5% sorbic acid. The diets were prepared with sterile water at a ratio 1.8 times the dry weight of the diet and heated at 121 °C for 30 min. Mulberry leaves (Yu-711) were obtained from the mulberry garden of the Sericulture Research Institute of Soochow University. The silkworm larvae were raised under standard conditions (26 ± 1 °C, 70 ± 5% humidity and a 12/12 h light/dark photoperiod). 

### 4.2. Physiological Index Measurements

The larvae of fourth instar silkworms were weighed at 0 h, 24 h and 48 h. Ten larvae from AD, ML and TM groups at 24 h and 48 h were dissected on ice and their intestinal fluid was taken for pH determination. Furthermore, ROS and antioxidant enzymes were measured in their midguts. The ROS assay kit was purchased from Nanjing Jiancheng Bioengineering Institute (Nanjing, China) and the antioxidant enzyme assay kits (SOD, TPX, CAT) were purchased from Sangon Biotech Co., Ltd. (Shanghai, China).

### 4.3. Body Weight Statistics and ROS Levels after E. cloacae Treatment

The bacterium *E. cloacae* (strain no. 1.2022) was obtained from the China Microbial Culture Collection (Beijing, China) and incubated in Luria–Bertani (LB) medium at 37 °C prior to adjustment to an optical density of 0.4 at 600 nm (OD_600_ = 0.4). The bacterial culture (50 mL) was pelleted via centrifugation (8000× *g*, 5 min), resuspended in 5 mL of sterile water, and then uniformly sprayed onto the feed surface. For the control group, the feed surface was sprayed with sterile water. After inoculation, body weight and ROS levels were measured in the silkworm. Three biological replicates were used for each assay.

### 4.4. DNA Preparation, PCR, Pyrophosphate Sequencing and Sequence Processing

Five larvae, each from the AD, ML and TM groups, were selected after 24 h and 48 h of instar 4, and 10 replicate samples were taken from each group. After disinfection of the body surface of the silkworm with 75% ethanol, the midgut was dissected and placed into a 2 mL sterile cryogenic vial on ice. Total nucleic acids were extracted using the E.Z.N.A. Soil DNA Kit (Omega Bio-Tek, Norcross, GA, USA). Meanwhile, total RNA was extracted according to the manufacturer’s instructions (Axygen Biosciences, Union City, CA, USA), and DNA-free RNA (0.5 mg/sample) was reverse-transcribed into cDNA.

PCR was performed in a 25 μL reaction system consisting of 0.8 μL of 5 mM forward primer 388F (5′-ACTCCTACGGGAGGCAGCAG-3′) and reverse primer 806R (5′-CGACTACHVGGGTWTCTAAT-3′), 2 μL of 2.5 mM dNTP, 0.4 μL of TransStart FastPfu DNA polymerase, 4 μL 5 × Fastpfu buffer (Takara, Dalian, China), 10 ng of template DNA and double-distilled water in a gradient Biometra 96 thermal cycler (Biometra, Göttingen, Germany). After an initial denaturation step at 94 °C for 3 min, amplification was performed with 25 cycles at a melting temperature of 95 °C of 30 s, annealing at 55 °C for 30 s and then extension at 72 °C for 10 min. Final amplicons were analyzed by electrophoresis on a 2% agarose gel (Biowest, Nuaillé, France), followed by staining with ethidium bromide and visualization under UV light. After a series of reactions, quantification, characterization and purification, samples were combined at equal concentrations and the 16S amplicons were pyrosequenced by Majorbio Bio-Pharm Technology Co., Ltd. (Shanghai, China).

### 4.5. Real-Time Quantitative Polymerase Chain Reaction

RT-qPCR was utilized to assess the transcript levels of genes associated with the Duox-ROS system, using Primer 6.0 to design specific primers (Primier Biosoft, Palo Alto, CA, USA) with the internal reference gene RP49 (Appendix A). RT-qPCR was performed using the ABI StepOnePlus TM Real-Time PCR system (Application Biosystems, San Francisco, CA, USA). The fluorescent dyes SYBR Premix ex Taq (Takara, Dalian, China) were employed for RT-qPCR with a total reaction volume of 20 μL, according to the manufacturer’s instructions. The relative expression levels were analyzed using the 2^−ΔΔCt^ method.

### 4.6. Sequencing Data Processing

The raw 16S rRNA gene sequencing reads were demultiplexed and paired-end reads were merged using FLASH (version 1.2.11 http://ccb.jhu.edu/software/FLASH/index.shtml (accessed on 10 March 2023.)) to combine the minimum 10 bp overlap with other default parameters. Initial sequencing data were processed using Microbial Ecology QIIME Quantitative Analysis (version 1.9.1 http://qiime.org/install/index.html (accessed on 14 March 2023)). Sequences with mismatched or unclear bases in the primer sequences were discarded, chimeras were removed, and valid sequences were clustered into operational taxonomic units (OTUs), with 97% sequence similarity using revealed USEARCH (http://www.drive5.com/usearch/ (accessed on 21 March 2023) version 11). Next, taxonomic identification was performed using the RDP classifier (version 11.5 https://sourceforge.net/projects/rdp-classifier/ (accessed on 7 April 2023)) and a reference dataset from the SILVA database (Release138 http://www.arb-silva.de (accessed on 18 April 2023)) (70% confidence threshold).

Mothur was utilized to calculate alpha diversity indices under random sampling and to determine alpha diversity and richness at the OTU level using the Shannon and Simpson indices, ACE and Chao1 indices, respectively; alpha diversity index calculation results are shown in Appendix A. The Wilcoxon rank sum test was used to compare statistical differences among groups, principal coordinate analysis (PCoA) was performed based on the Bray–Curtis dissimilarity matrix calculated from the samples, and non-metric multidimensional scaling (NMDS) was used to visualize the phylogenetic distance between bacterial communities of different samples. *p* < 0.05 (*) was considered statistically significant. The Majorbio cloud platform was utilized to perform the analyses (www.majorbio.com (accessed on 9 March 2023)). Statistical analyses and plotting were performed using R software (version 3.3.1) and graphs were generated using GraphPad Prism 8 (GraphPad, San Diego, CA, USA).

### 4.7. Statistical Analysis

Body weight, pH and enzyme activity of silkworms were expressed as the mean with standard error (SE) using one-way analysis of variance (ANOVA) to compare values among groups. Statistical analysis was performed using SPSS 26.0 (SPSS, Chicago, IL, USA) for the Bonferroni method. Statistical significance was set at *p* < 0.05 (*).

## 5. Conclusions

After feeding artificial diets to silkworms, their body weight and immune capability decreased, and their intestinal microbiota tended to be homogeneous. After resuming mulberry leaf feeding, silkworms experienced a period of adaptation, and their intestinal pH gradually increased to facilitate the screening of adaptive intestinal microorganisms. This experiment represents a systematic study of the gut of the silkworm from the microbiological point of view, and elucidated differences in the intestinal microbiota induced by three distinct feeding methods. Furthermore, the adaptive mechanisms are hypothesized above. Herein, the following results were observed: (1) Traditional mulberry leaf rearing of silkworm resulted in the presence of intestinal microbiota that was rich and stable; silkworms’ body weight and immune performance were optimal. Meanwhile, artificial feed rearing will result in an acidic intestinal environment in silkworms, affecting both the microbiota and the physical and chemical properties of the intestinal fluid, further causing poor nutrient absorption and weak immune resistance to disease. (2) After being transferred from artificial feeding to mulberry leaf feeding, it was identified that the growth and immunity of silkworms could return to normal. (3) The artificially fed silkworms exhibited a low immune tolerance, low ROS content and high antioxidant enzyme content, but long periods of time without replacing the artificial feed would lead to the production of pathogenic bacteria that would be detrimental to the growth of silkworms. (4) The study established that food–gut microbiota–Duox defense system interactions may provide a reference for studying new functions of insect gut microbes.

## Figures and Tables

**Figure 1 ijms-24-12731-f001:**
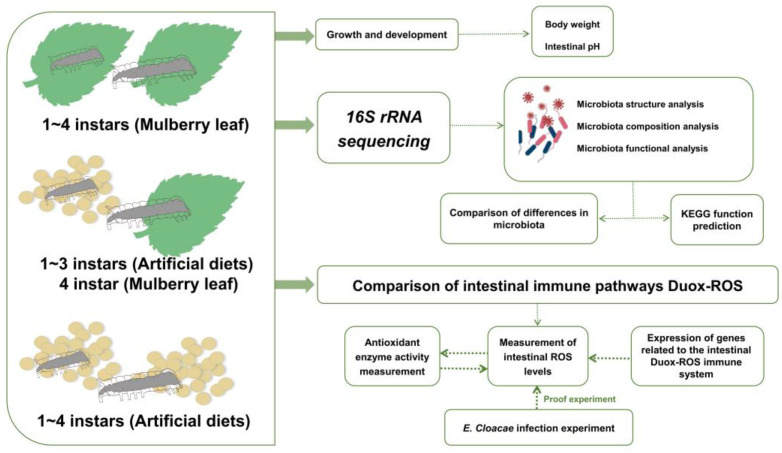
Flow chart of the experiment. Three different treatment groups were fed to silkworms: 1–4 instars artificial diet (AD), 1–4 instars mulberry leaf (ML) and 1–3 instars artificial diet + 4 instar mulberry leaf (TM). 16S rRNA sequencing was used to compare the changing characteristics of the microorganisms of the intestinal microbiota of silkworms in relation to interactions with the intestinal immune system, DUOX-ROS.

**Figure 2 ijms-24-12731-f002:**
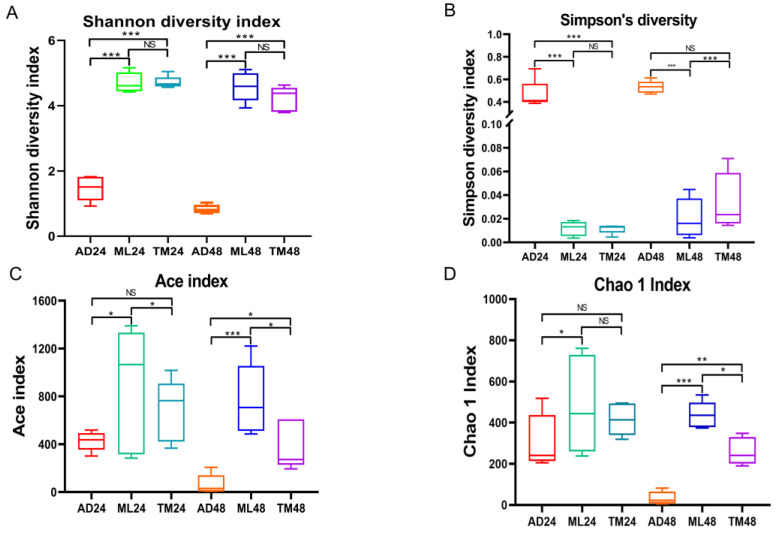
α-Diversity of the gut microbiota of *B. mori*. (**A**) Shannon index. (**B**) Simpson index. (**C**) Chao1 index. (**D**) Ace index (*n* = 5 in each group). *p* < 0.05 (*); *p* < 0.01 (**); *p* < 0.001 (***); NS, not significant by Student’s *t*-test.

**Figure 3 ijms-24-12731-f003:**
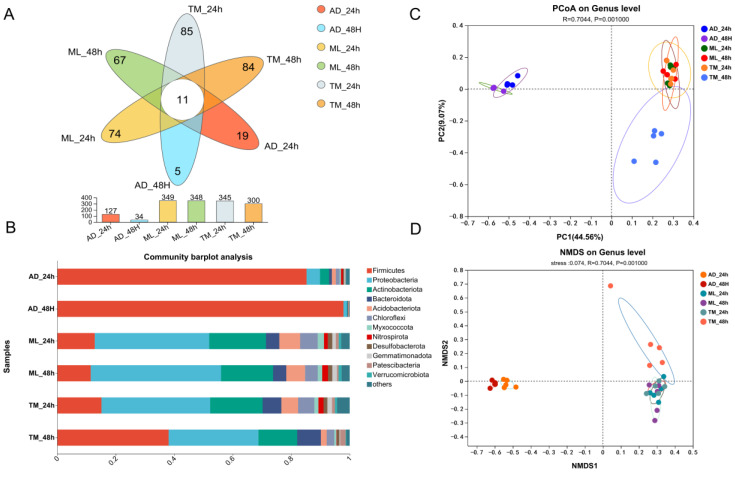
Dynamics of the gut microbial community of the *B. mori*. (**A**) Venn diagram displaying the number of OTUs distributed. (**B**) Relative abundance of bacterial phyla level in different samples. (**C**) Visualization of gut bacterial community clusters in *B. mori* on PCoA plots (97% similarity level) based on Bray–Curtis distances (ANOSIM, *p* = 0.001). (**D**) Visualization of gut bacterial community clusters in *B. mori* on NMDS plots (97% similarity level) based on Bray–Curtis distances (ANOSIM, *p* = 0.001) (*n* = 5 in each group).

**Figure 4 ijms-24-12731-f004:**
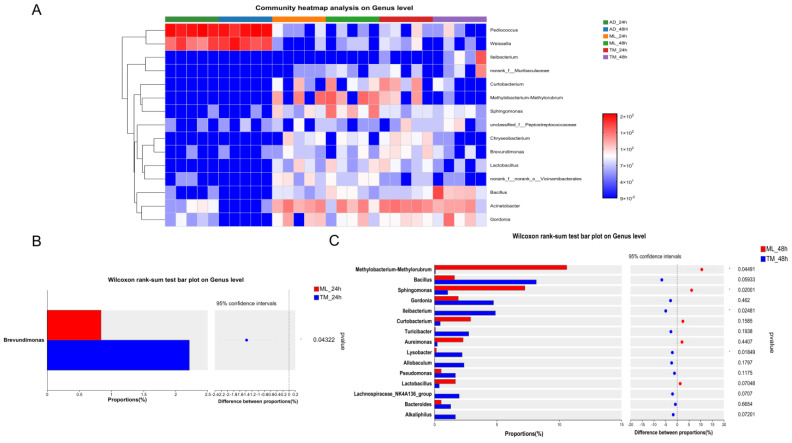
Differences in the composition of the intestinal microbiota of the *B. mori*. (**A**) Heat map illustrating the relative abundance of dominant bacterial genera. Each column represents a silkworm larva. (**B**) Differences in microbiota between the TM and ML groups at 24 h. (**C**) Differences in microbiota between the TM and ML groups at 48 h.

**Figure 5 ijms-24-12731-f005:**
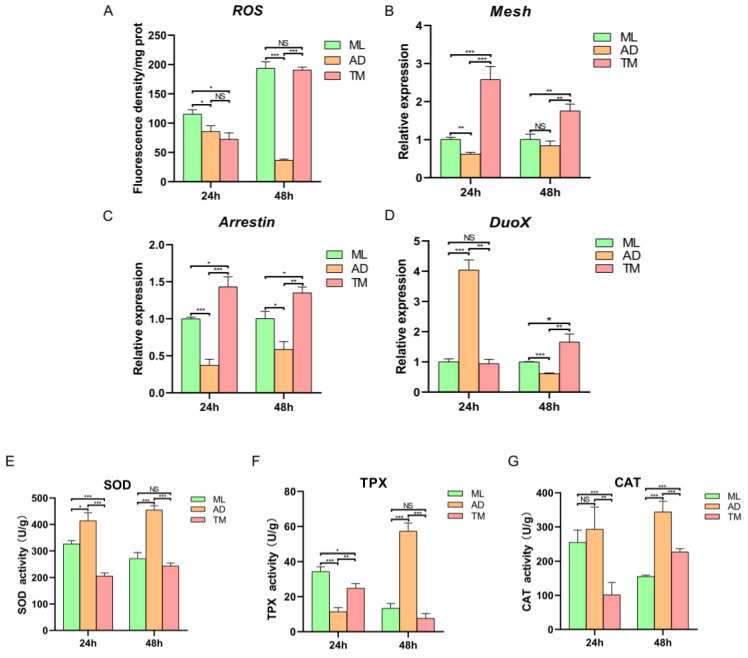
ROS-level detection and antioxidant enzyme activity. (**A**) ROS-level detection. (**B**) *Mesh*, (**C**) *Arrestin* and (**D**) *Duox*; Duox-ROS-related genes; (**E**) SOD, (**F**) TPX and (**G**) CAT; antioxidant enzyme activity. The results are indicated as mean ± SE (*n* = 5). *p* < 0.05 (*); *p* < 0.01 (**); *p* < 0.001 (***); NS, Not significant by Student’s *t*-test.

**Figure 6 ijms-24-12731-f006:**
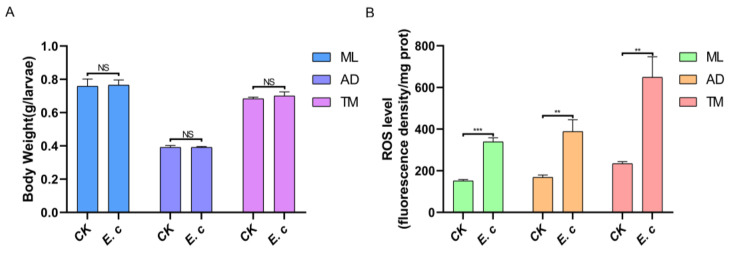
Body weight and ROS levels of silkworms fed with *E. cloacae*. (**A**) Weight of larvae of the silkworm. (**B**) ROS-level detection. *p* < 0.01 (**); *p* < 0.001 (***); NS, not significant by Student’s *t*-test.

## Data Availability

The data presented in this study are available in the Appendix A, 16S microbiomes raw sequencing raw sequencing data were deposited in the National Center for Biotechnology Information, BioProject PRJNA999036, under the accession numbers SRP451657.

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
