# Peer review of "Homeostatic Regulation of the Duox-ROS Defense System: Revelations Based on the Diversity of Gut Bacteria in Silkworms (Bombyx mori)"

_ijms, 2023, doi:10.3390/ijms241612731_

Round 1
Reviewer 1 Report
The paper focuses on the interactions that occur between gut microbiota of Bombyx mori and host intestinal dual oxidase (DUOX) system. It contains interesting results, which, however, are poorly described and discussed. The language style needs significant improvement. First of all, the authors should not use the term “intestinal flora” or “bacterial flora”, because these are obsolete expressions that are not currently used in the scientific literature. Instead, it is recommended to use the term “microbiota”. Authors often forget to use italics in the names of bacterial species or genera, e.g. L87, L110, L118.
In the Introduction the authors do not provide enough information about how diet could affect the composition of the intestinal microbiota of B. mori. I would also suggest the authors should improve the selection of literature references, especially at the beginning of the Introduction section.
In the Results section, many statements are unclear, only the figures show what results were obtained, e.g. L79-85. The quality of the figures and tables are generally satisfactory, although figures 2A, 2B, 2C are too small and hard to read. The authors refer to important results regarding the pH of the intestinal fluid in the discussion, but do not include any data on this subject in the Results section. I also did not find any charts or tables on this subject in the supplementary data that the authors refer to. The Supplementary materials subsection is left blank (L 95). I also couldn't find the rarefraction curves for the obtained reads, that should have been included. Additionally, the obtained metagenomics data are missing at the NCBI database. The authors should add the metagenomics to the created Bioproject and Biosamples. The results of the discussion should be better discussed, not just presented, especially in subsection 3.1. and 3.2
Other comments:
L3 - gut colonies – “gut bacteria” would be better
L16, 209 - single intestinal flora – should be” reduced” or “simple intestinal microbiota”
L18 - and microorganisms – should be “microorganism composition”
L68, 392 - gut microbial– should be “gut microbiota”
L95- lowest OTU –should be “lowest number of OTU”
L95 – and demonstrated a decreasing trend, with the same groups 11 - this part of sentence is unclear
L97 – which increased to 97,86% - Initial value of abundance should be provided – increased from …. to 97,86%
L100 – the phylum level - should be “bacterial composition at the phylum level”
L104 – dispersed? – should be rather “separate”
L116 – microorganisms in the silkworm – “microorganisms at the genus level” should be added
L155-159 - such information fit more closely to the Discussion section.
The English language of the paper is not sufficient quality and needs improvement.
Author Response
Manuscript ID IJMS-2520197 entitled " Homeostatic regulation of the Duox-ROS defense system: rev-elations based on the diversity of gut bacteria in the silkworm (Bombyx mori)" which I have submitted to International Journal of Molecular Sciences, has been reviewed. The comments of the referees are included at the letter.
Response to Reviewer 1 Comments
Major comments
Point 1: The paper focuses on the interactions that occur between gut microbiota of Bombyx mori and host intestinal dual oxidase (DUOX) system. It contains interesting results, which, however, are poorly described and discussed. The language style needs significant improvement. First of all, the authors should not use the term “intestinal flora” or “bacterial flora”, because these are obsolete expressions that are not currently used in the scientific literature. Instead, it is recommended to use the term “microbiota”. Authors often forget to use italics in the names of bacterial species or genera, e.g. L87, L110, L118.
Response: Thank you for your comment. We have made the changes accordingly in the revised manuscript. We have made some modifications to the results and discussion sections, and the language has been corrected in that we have used microbiota and used italics for species and genera in bacteria.
Point 2: In the Introduction the authors do not provide enough information about how diet could affect the composition of the intestinal microbiota of B. mori. I would also suggest the authors should improve the selection of literature references, especially at the beginning of the Introduction section.
Response: Thank you for your comment. Because the silkworm mainly feeds on mulberry leaves, by adding other substances in mulberry leaves, which can achieve to change the composition of the intestinal flora of the silkworm, we add the related literature, and found that low concentration of TiO2 NPs changed the abundance of the genus of the silkworm's microbiota, and promoted the growth of the silkworm; We have re-selected the references at the beginning of the Introduction section.
Point 3: In the Results section, many statements are unclear, only the figures show what results were obtained, e.g. L79-85. The quality of the figures and tables are generally satisfactory, although figures 2A, 2B, 2C are too small and hard to read. The authors refer to important results regarding the pH of the intestinal fluid in the discussion, but do not include any data on this subject in the Results section. I also did not find any charts or tables on this subject in the supplementary data that the authors refer to. The Supplementary materials subsection is left blank (L 95). I also couldn't find the rarefaction curves for the obtained reads, that should have been included. Additionally, the obtained metagenomics data are missing at the NCBI database. The authors should add the metagenomics to the created Bioproject and Biosamples. The results of the discussion should be better discussed, not just presented, especially in subsection 3.1. and 3.2.
Response: Thank you for your comment. In section L79-85, we placed the specific data in the Supplementary materials Table S1. We corrected the graphs in the article for easier reading and re-uploaded the Supplementary materials; The pH was placed in the Supplementary materials Fig. S1. and the Rarefaction curves in the Supplementary materials Fig. S2. The 16S microbiomes raw sequencing data were deposited in the National Center for Biotechnology Information, BioProject PRJNA999036, under the accession numbers SRP451657; In the discussion section, we have made the changes accordingly in the revised manuscript.
Minor comments
Point 1: L3 - gut colonies – “gut bacteria” would be better.
Response: Thank you for your comment. We have made the changes accordingly in the revised manuscript.
Point 2: L16, 209 - single intestinal flora – should be” reduced” or “simple intestinal microbiota”.
Response: Thank you for your comment. We have made the changes accordingly in the revised manuscript.
Point 3: L18 - and microorganisms – should be “microorganism composition”.
Response: Thank you for your comment. We have made the changes accordingly in the revised manuscript.
Point 4: L68, 392 - gut microbial– should be “gut microbiota”.
Response: Thank you for your comment. We have made the changes accordingly in the revised manuscript.
Point 5: L95- lowest OTU –should be “lowest number of OTU”.
Response: Thank you for your comment. We have made the changes accordingly in the revised manuscript.
Point 6: L95 – and demonstrated a decreasing trend, with the same groups 11 - this part of sentence is unclear.
Response: Thank you for your comment. We have made the changes accordingly in the revised manuscript.
Point 7: L97 – which increased to 97,86% - Initial value of abundance should be provided – increased from …. to 97,86%.
Response: Thank you for your comment. We have made the changes accordingly in the revised manuscript.
Point 8: L100 – the phylum level - should be “bacterial composition at the phylum level”.
Response: Thank you for your comment. We have made the changes accordingly in the revised manuscript.
Point 9: L104 – dispersed? – should be rather “separate”.
Response: Thank you for your comment. We have made the changes accordingly in the revised manuscript.
Point 10: L116 – microorganisms in the silkworm – “microorganisms at the genus level” should be added.
Response: Thank you for your comment. We have made the changes accordingly in the revised manuscript.
Point 11: L155-159 - such information fit more closely to the Discussion section.
Response: Thank you for your comment. We have deleted this section.

Reviewer 2 Report
The manuscript by QiLong Shu and co-authors entitled "Homeostatic regulation of the Duox-ROS defense system: revelations based on the diversity of gut colonies in the silkworm (Bombyx mori)" is devoted to the role of the Duox-ROS defense system depending on the three different feeding methods. The results look promising, but the manuscript has few flaws.
I did not find that the raw sequencing data obtained from 16S microbiomes were deposited in a public repository. It is very important for verification and reproducibility purposes.
Another major issue, I did not find any supplementary material mentioned in the paper (Figures S and Tables S). I was not attached to the manuscript files.
Also, I suggest to check the manuscript and italicize Latin nomenclature names everywhere (e.g. lines 45, 56, 87, 97, 99, 100, etc...).
Other issues include:
L005: What is affilation #3?
L010: Omit an extra dot.
Fig. 1: I can't find any reference to Figs. 1C and 1D in the manuscript text.
L107: What is NMDS?
Figure 3A: The legend text is illegible.
Line 148: Duox (L175) or DuoX (Fig. 4D)? Also, italicize gene names throughout the manuscript.
Figure 4: Too many white spaces. Can you make the figures larger?
L245: Shorten Enterobacter to E.
In general, the English is okay, but there are some parts that I can't understand.
For example:
I cant understand what means "The intestinal flora of silkworm in the AD group was single".
Author Response
我已提交给《国际分子科学杂志》的手稿ID IJMS-2520197题为“Duox-ROS防御系统的稳态调节:基于蚕(Bombyx mori)肠道细菌多样性的转速”。推荐人的评论包含在信中。
Manuscript ID IJMS-2520197 entitled " Homeostatic regulation of the Duox-ROS defense system: rev-elations based on the diversity of gut bacteria in the silkworm (Bombyx mori)" which I have submitted to International Journal of Molecular Sciences, has been reviewed. The comments of the referees are included at the letter.
Response to Reviewer 1 Comments
Major comments
Point 1: I did not find that the raw sequencing data obtained from 16S microbiomes were deposited in a public repository. It is very important for verification and reproducibility purposes.
Response: Thank you for your comment. We've uploaded the 16S microbiomes raw sequencing data were deposited in the National Center for Biotechnology Information, BioProject PRJNA999036, under the accession numbers SRP451657.
Point 2: Another major issue, I did not find any supplementary material mentioned in the paper (Figures S and Tables S). I was not attached to the manuscript files.
Response: Thank you for your comment. We will re-upload the supplementary material S1 and Tables S1.
Point 3: Also, I suggest to check the manuscript and italicize Latin nomenclature names everywhere (e.g. lines 45, 56, 87, 97, 99, 100, etc...).
Response: Thank you for your comment. We have made the changes accordingly in the revised manuscript.
Minor comments
Point 1: L005: What is affilation #3?
Response: Thank you for your comment. We have made the changes accordingly in the revised manuscript.
Point 2: L010: Omit an extra dot.
Response: Thank you for your comment. We have made the changes accordingly in the revised manuscript.
Point 3: Fig. 1: I can't find any reference to Figs. 1C and 1D in the manuscript text.
Response: Thank you for your comment. We have added it to the manuscript.
Point 4: L107: What is NMDS?
Response: Thank you for your comment. non-metric multidimensional scaling (NMDS) is a data analysis method that simplifies research objects (samples or variables) in multidimensional space to a low-dimensional space for localization, analysis, and categorization, while preserving the original relationships between objects. Its basic feature is to view the similarity or dissimilarity data between objects as a monotonic function of the distance between points, and to replace the original data with new columns of the same order on the basis of maintaining the original data order relationship for metric multidimensional scaling analysis. NMDS is often used for β-diversity analysis of microorganisms.
Point 5: Figure 3A: The legend text is illegible.
Response: Thank you for your comment. We have made the changes accordingly in the revised manuscript.
Point 6: Line 148: Duox (L175) or DuoX (Fig. 4D)? Also, italicize gene names throughout the manuscript.
Response: Thank you for your comment. We have made the changes accordingly in the revised manuscript.
Point 7: Figure 4: Too many white spaces. Can you make the figures larger?
Response: Thank you for your comment. We have enlarged the figures as you requested.
Point 8: L245: Shorten Enterobacter to E.
Response: Thank you for your comment. We have made the changes accordingly in the revised manuscript.

Round 2
Reviewer 2 Report
All comments have been taken into account and the manuscript is ready to be accepted.
English quality is okay.
Author Response
We thank the reviewers for their comments and revisions of our manuscript.